

# Formulating and testing a method for perturbing precipitation time series to reflect anticipated climatic changes

Hjalte Jomo Danielsen Sørup[1,2], Stylianos Georgiadis[1,3], Ida Bülow Gregersen[4] and Karsten Arnbjerg-Nielsen[1,2]

[1]Technical University of Denmark, Global Decision Support Initiative, Lyngby, Denmark
[2]Tecnical University of Denmark, Department of Environmental Engineering, Lyngby, Denmark
[3]Tecnical University of Denmark, Department of Applied Mathematics and Computer Science, Lyngby, Denmark
[4]Ramboll Danmark A/S, Department of Climate Adaptation and Green Infrastructure, Copenhagen, Denmark

*Correspondence to*: Hjalte J.D. Sørup (hjds@env.dtu.dk)

**Abstract.** Urban water infrastructure has very long planning horizons and planning is thus very dependent on reliable estimates on the impacts of climate change. Many urban water systems are designed using time series with a high temporal resolution. To assess the impact of climate change on these systems similarly high resolution precipitation time series for future climate are necessary. Climate models cannot at their current resolutions provide these time series at the relevant scales. Known methods for stochastic downscaling of climate change to urban hydrological scales have known shortcomings

in constructing realistic climate changed precipitation time series at the sub-hourly scale. In the present study we present a deterministic methodology to perturb historical precipitation time series at minute scale to reflect non-linear expectations to climate change. The methodology shows good skill in meeting the expectations to climate change of extremes at event scale when evaluated at different timescales from the minute to the daily scale. The methodology also shows good skill with respect to representing expected changes to seasonal precipitation. The methodology is very robust to the actual magnitude

of the expected changes as well as the direction of the changes (increase/decrease) even for situations where the extremes are increasing for seasons that in general should have a decreasing trend in precipitation. The methodology can provide planners with valuable time series representing future climate that can be used as input to urban hydrological models and give better estimates of climate change impacts on these systems.

## 1 Introduction

Climate change impact water management worldwide as the water cycle is an essential part of the climate system. The planning horizon for water infrastructure is often very long, making reliable predictions of future climate crucial (Arnbjerg-Nielsen et al., 2015b). In design of water infrastructure precipitation data is needed. Especially for urban infrastructure the time resolution of precipitation data needed for design and planning is much finer than what is provided by climate models (Berndtsson and Niemczynowicz, 1988; Schilling, 1991). Hence a lot of effort is put into giving reliable estimates of what

the expected change in precipitation will be at these fine scales (Fowler et al., 2007; Kendon et al., 2014; Mayer et al., 2015). Expected changes in precipitation, however, does not translate directly into changes in floods or overflows from structures. To determine these changes, urban hydrological models have to be run driven by the changed precipitation (Olsson et al.,




2009; Willems et al., 2012). By definition fine resolution precipitation time series for future climates are not observable and hence a multitude of statistical approaches have been developed to enable generation of time series with properties that for a large range of metrics have the same characteristics as the expected future precipitation (Willems, 1999; Olsson and Burlando, 2002; Cowpertwait, 2006; Molnar and Burlando, 2008; Burton et al., 2010; Willems et al., 2012; Sørup et al., 2016a).

Expectations to precipitation at event level under climate change are often non-linear with the anticipation that changes in occurrence and size of extreme events are higher than changes in seasonal or yearly precipitation (Boberg et al., 2010). This is a problem often sought solved by weather generators or other similar downscaling techniques (Fowler et al., 2007; Burton et al., 2010) but these often have difficulty in presenting realistic time series at the sub-hourly to hourly time scales relevant for urban infrastructure (Segond et al., 2006; Verhoest et al., 2010; Sørup et al., 2016a).

In the present study we develop and demonstrate a novel non-linear methodology that perturbs existing precipitation time series to reflect complex expectations to precipitation in a changed future climate. The method incorporates regional historical knowledge about precipitation through the use of Intensity-Frequency-Duration (IDF) relationships (WMO, 2009) and knowledge about the expected changes of these due to climate change. Thus, the method generates time series for a changed climate which are chronologically identical to the observations used as input but perturbed to reflect climate change. These series can be used as input for hydraulic or hydrologic models where the climate change effect has to be assessed for all possible rain conditions.

The presented methodology is based on the assumption that precipitation can be scaled according to identified expectations to climate changes. It is rather similar to Distribution Based Scaling presented by Yang et al. (2010). Unlike the study by Yang et al. (2010) the expected changes are not calculated directly using the Delta Change method (Fowler et al., 2007) on Regional Climate Model output; they are derived from comprehensive state-of-the-art studies where the full available data basis is used to determine realistic expectations to changes to precipitation due to climate change (e.g. Giorgi, 2006; Kendon et al., 2008; Christensen et al., 2015).

## 2 Methodology

In urban water management, the relevant time frame to consider is most often that of the rain event (Willems, 1999). The determination of robust IDF relationships for present climate at the relevant time scales is a prerequisite. For developed countries where high resolution precipitation is generally available these two prerequisites are very often met (Arnbjerg-Nielsen et al., 2015a), making the methodology generally relevant.





### 2.1 Modelling Framework

Let us consider a system $S$ that describes precipitation over a time period. The original data are expressed as a time series of precipitation intensity over fixed time steps. This time series alternates between a dry period (no precipitation) and a rainy one. A given event is characterized as dry, *extreme* or *non-extreme* with respect to amount of precipitation during the event.

We denote by $E$, $|E| \geq 3$, the state space of the system $S$. Let also $D_0$, $D_1$ and $D_2$ be the non-empty sets of states of dry periods, non-extreme and extreme events, respectively, with $|D_0| = 1$, $|D_1| = d_1 \geq 1$ and $|D_2| = d_2 \geq 1$, i.e. there exists exactly one state for dry periods, $D_0 = D^{dry}$, but several different states ($d_1$ and $d_2$; respectively), can be defined for both the non-extreme and extreme events. An extreme event can be further categorized according the severity of the phenomenon, expressed in terms of the return period of the measured intensity. Non-extreme events can be categorized according to the

season in which they appear. Hence, the state space $E$ is partitioned into three disjoint subsets as follows:

$$E = D_0 \cup D_1 \cup D_2, \tag{Eq. 1}$$

where $D_i \cap D_j = \emptyset$, $i \neq j$, $i,j \in \{0,1,2\}$. We link the non-extreme events to the seasonality of the phenomenon and thus $D_1 = \{D^{winter}, D^{spring}, D^{summer}, D^{fall}\}$, that is $d_1 = 4$. $D_2$ can likewise be partitioned into one or several states appropriate to describe extreme precipitation which may have different return periods or different hydro-climatic origin. In this study we

use a partition based on return periods with $D_2 = \{D^2, D^{10}, D^{100}\}$, referring to states that classifies the extremes as either 2, 10 and 100-year events based on return level.

By definition there is always a dry period between two events and we assume that the there is no dependence between consecutive events. We define the following processes that describe the evolution of a semi-Markov system (Barbu and Limnios, 2008):

•   $\mathcal{J} := (\mathcal{J}_n)_{n \in N}$ is a Markov chain with state space $E$, where $\mathcal{J}_n$ is the state of the system at the *n*-th event;

•   $U := (U_n)_{n \in N}$ is the sequence of jump times between states with state space $N$ and $U_0 = 0$;

•   $Z := (Z_k)_{k \in N}$ is a discrete-time process with states on $E$, with $Z_k$ to be the state of the system at a time step $k$.

The processes $\mathcal{J}$ and $Z$ are related through the formula

$$Z_k = \mathcal{J}_{N(k)}, k \in N, \tag{Eq. 2}$$

where $N(k)$ is the discrete-time counting process of events in $[1, k] \subset N$, i.e.

$$N(k) := max\{n \in N : U_n \leq k\}. \tag{Eq. 3}$$

The corresponding transition matrix of the chain $\mathcal{J}$ is very simple to be written. Figure 1b illustrates the evolution of the

stochastic system described above.





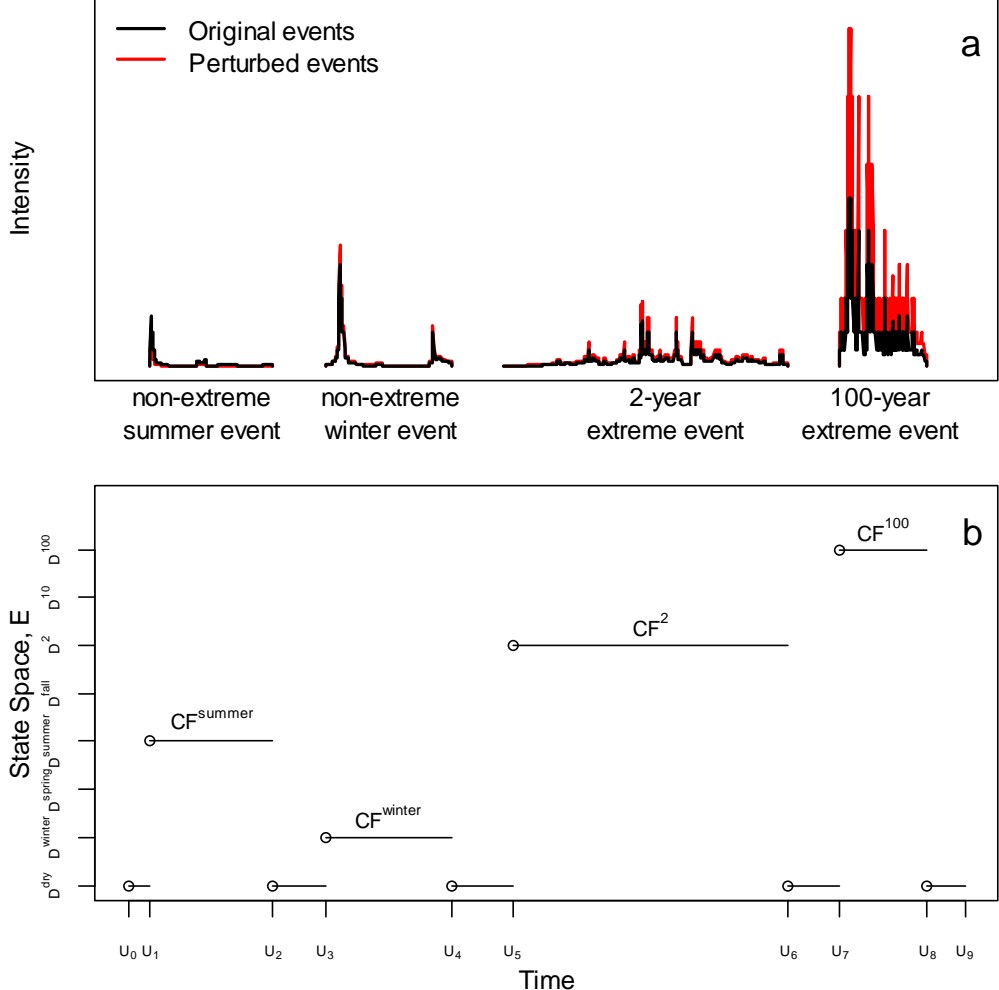

**Figure 1 a) Illustration of the magnitude of perturbation of events for non-extreme summer and winter events as well as 2- and 100-year extreme events, with summer events being perturbed with a factor below one and factors for the winter and the extremes being above one. Factors for extremes are higher than for the winter events, and factors for the very extreme is higher than for the more moderate extreme. b) Illustration of the states associated with the different events if they were to happen in the shown chronology, the dry state, $D^{dry}$, is present between all wet states.**





### 2.2 Framework for determining state of individual events

There is no unique way to assign a state to an extreme event. In this paper, various methods based on the maximum mean intensities are used to define the event state. For all investigated methods the changes to extremes are evaluated by calculation of IDF curves at event level (WMO, 2009). The return period of individual events across all intensities is

determined using the median plotting position (Rosbjerg, 1988):

$$T_{median} = \frac{T_{total}+0.4}{rank-0.3},$$  (Eq. 4)

where $T_{total}$ is the length of the time series and *rank* is the rank number of the individual event.

Using data with observations every minute and a minimum dry weather separation between of 60 minutes, the mean maximum intensities over 5, 10, 30, 60, 180, 360 and 720 minutes are calculated for each event. At shorter timeframes, e.g.

one minute, the variability of the observed extremes are expected to be very large due to the inherent sampling error (Fankhauser, 1998) and at very long timeframes, e.g. one day (1440) the extremes are often consisting of several events following one another and a different event definition would be necessary to ensure that the real extremes is identified (Madsen et al., 2009). A representative return period for the event is derived based on a mathematical comparison to regional IDF estimates (Figure 2). This return period is then in turn used to define the state of the event. We test four different

selection criteria which define the state of extreme events as either $D^2$, $D^{10}$ or $D^{100}$. The selection criteria are listed in Section 3.3.

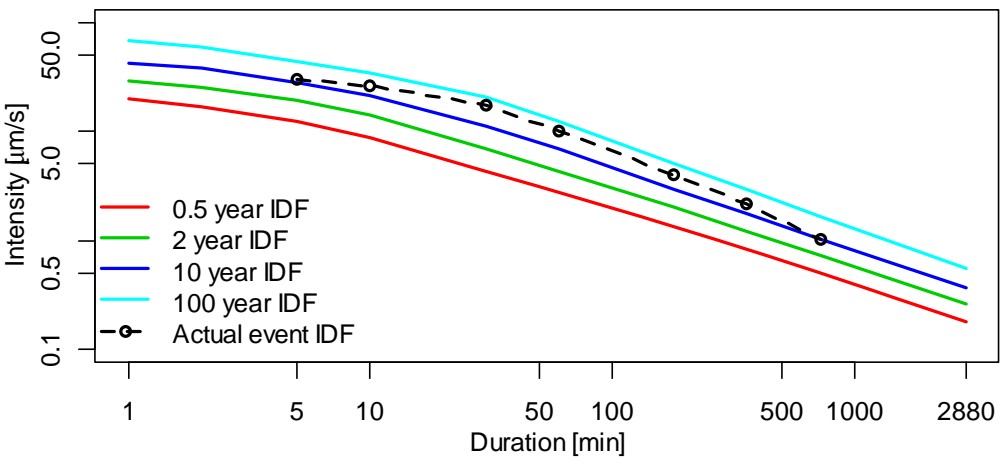

**Figure 2 The IDF curve for an extreme event in comparison to the regional IDF curves for 0.5, 2, 10 and 100 year return periods respectively (based on Madsen et al. (submitted)).**






### 2.3 Perturbation and change factor

With each event of a time series classified according to a state, the time series can be perturbed using the following methodology linking the time series to the states of the individual events.

Let $R_k$, $k \in N$, be the precipitation intensity at time step $k$ and $R := (R_k)_{k \in N}$ the corresponding process describing these
intensities. The process of perturbed precipitation in each time step $k$ is denoted by $R^* := (R_k^*)_{k \in N}$.

Similarly to the state space $E$, we introduce the state space of the change factors, denoted by $E_{CF}$, $|E_{CF}| = |E|$. We can then write

$$E_{CF} = C_0 \cup C_1 \cup C_2, \tag{Eq. 5}$$

with $|C_0| = 1$, $|C_1| = d_1$ and $|C_2| = d_2$.

We consider the process $CF := (CF_n)_{n \in N}$ with state space $E_{CF}$, where $CF_n$ is the change factor at the $n$-th event. Let $W := (W_k)_{k \in N}$ be the chain, with state space $E_{CF}$, of change factors in time steps $k \in N$, that is

$$W_k = CF_{N(k)}, \tag{Eq. 6}$$

with $N(k)$ to be the counting process defined in (Eq. 3). Under the above notation, the original and perturbed sequences of precipitation, $R_k$ and $R_k^*$, are written as

$$R_k^* = W_k R_k. \tag{Eq. 7}$$

This means that for a sequence of events some events will be perturbed more than others and for extreme cases some might be reduced while others are increased depending on the local expectations to climate change. Figure 1a shows an example where a non-extreme summer event is perturbed to a lesser volume than original while a winter non-extreme is increased marginally and both 2 and 100 year extremes are increased considerably more (both in absolute numbers as well as in
relative percentages). Figure 1b shows how the state space changes if these four events were to happen chronologically in time with the state jump times marked at the x-axis.

### 2.4 Volume correction based on seasonal dependence of extremes

The extreme part of precipitation is only expected to constitute a smaller fraction of the total precipitation volume on an
annual basis (Sørup et al., 2016b) but as extreme precipitation is often associated with a particular season (see e.g. Sørup et al., 2012) the volumetric part of the extremes might be higher for sub-annual considerations. This implies that situations like "increased global precipitation is likely to be experienced as heavier precipitation events, rather than an increase in the frequency of precipitation" (Fowler and Hennessy, 1995) have to be handled through volumetric corrections, to accommodate that both expectations to changes in extremes and overall seasonal changes are correct. How to do this best
will be very much dependent on the local conditions. In our case this is described in Section 3.4.





## 2.5 Evaluation of Perturbed Time Series

The evaluation of the perturbed time series is done against the original time series and against the expected changes.

The average percentwise difference between the perturbed return levels, $z^*_{i,j,m}$, of the modelled time series, $R^*_k$, perturbed

with the time dependent change factors, $W_k$, against the same return levels, $z_{i,j,m}$, of the original time series, $R_k$, multiplied

with the target change factor, $CF^j_e$, can be defined as:

$$\Phi_{i,j,m} = \left(1 - \frac{z^*_{i,j,m}}{z_{i,j,m}CF^j_e}\right)100\% \, , \qquad\qquad \text{(Eq. 8)}$$

across all IDF points, $i$, all extremity levels and seasonality, $j$, and all perturbed time series, $m$. A combined skill score, $\Phi$,

across all considered metrics that describe the average deviance from the expectations can then be defined as:

$$\Phi = \sum_{i \in I} \sum_{j \in J} \sum_{m \in M} \frac{|\Phi_{i,j,m}|}{|I|\,|J|\,|M|} \, , \qquad\qquad \text{(Eq. 9)}$$

With $|I|\,|J|\,|M|$ being the product of the total number of IDF points, $I$, the total number of extreme levels considered plus

seasonality, $J$, and the total number of time series perturbed , $M$, as a normalization factor.

## 2.6 Sensitivity Analysis

The robustness of the methodology is tested by evaluating its sensitivity to the actual magnitude of the target parameters for

both extreme and seasonal changes. Low, mean and high scenarios are constructed and paired in all possible combinations to

assess both the individual and combined influence of these (Table 1). As this increases the number of scenarios with which

to perturb the precipitation time series substantially, this is not done until after an initial evaluation of the state selection

criteria.

**Table 1 Tested combinations of extreme and seasonal changes.**

| Seasonality | Extremes | | |
|---|---|---|---|
| | Low expected change | Mean expected change | High expected change |
| **Low expected change** | LL | ML | HL |
| **Mean expected change** | LM | MM | HM |
| **High expected change** | LH | MH | HH |


## 3 Case study: Denmark

To showcase the methodology it is applied to Danish conditions where the situation is that complex non-linear changes are

expected with respect to precipitation in a changed climate.





### 3.1  Data

### 3.1.1 Observational Data

Precipitation data from the Danish SVK rain gauge network is used this study (Mikkelsen et al., 1998; Madsen et al., 2002).
For this study 10 time series from different parts of Denmark with lengths of approximately 33 years between 1979 and 2012

5   are used. To distinguish individual events a dry weather period between individual events of at least 60 minutes is applied.

### 3.1.2    IDF Curves

For present climate IDF curves are extracted from a regional model for extremes originally developed by Madsen et al.
(1998) and updated by Madsen et al. (2009, submitted). The IDF curves vary across Denmark but a single mean regional
curve is chosen for this study independent of the location of the gauge considered. Table 1 summarizes the IDF values used.

**Table 2 IDF values for various return periods for Denmark extracted from the model presented by Madsen et al. (submitted).**

| Return Period (years) | Intensities (µm/s) | | | | | | |
| --- | --- | --- | --- | --- | --- | --- | --- |
| | Duration (min) | | | | | | |
| | 5 | 10 | 30 | 60 | 180 | 360 | 720 |
| T=100 | 43.67 | 34.80 | 20.63 | 12.47 | 5.21 | 3.11 | 1.72 |
| T=10 | 28.62 | 21.43 | 11.37 | 6.95 | 3.09 | 1.86 | 1.09 |
| T=2 | 19.54 | 14.08 | 7.08 | 4.38 | 2.04 | 1.25 | 0.75 |
| T=0.5 | 12.40 | 8.73 | 4.33 | 2.75 | 1.33 | 0.84 | 0.51 |

### 3.1.3    Expectations to Climate Change

The official recommendations regarding climate change for urban infrastructure in Denmark was determined by Gregersen et

15   al. (2014) on the basis of the ENSEMBLES data set (van der Linden and Mitchell, 2009), with the addition of a few
simulations using high-end scenarios. Table 2 sums up these expectations.

20





**Table 3 Expected changes in extreme precipitation for Denmark. All values from Gregersen et al. (2014).**

| Change factor for extreme precipitation (-) | 2 year event [CF$^2$] | 10 year event [CF$^{10}$] | 100 year event [CF$^{100}$] |
| --- | --- | --- | --- |
| **Low expected change** | 1.0 | 1.0 | 1.0 |
| **Mean expected change** | 1.2 | 1.3 | 1.4 |
| **High expected change** | 1.45 | 1.7 | 2.0 |

In addition the Danish Meteorological Institute has published expectations regarding the change in precipitation on a seasonal basis (Table 4). Olesen et al. (2014) estimated these change factors based on analysis of the RCP2.6 and the RCP8.5

5   scenarios (Moss et al., 2010), hence, a low-and a high-end emission scenario, respectively. To match the change factors for extreme precipitation in Gregersen et al. (2014), which primarily is based on the average emission A1B scenario (Nakicenovic et al., 2000), simple scaling of the seasonal expectations to a mid-point is applied, as scalability has been shown to be a valid assumption across most scales and most indices (Christensen et al., 2015). The original estimates from Olesen et al. (2014) are kept as low and high expected changes for the sensitivity analysis.

**Table 4 Expected seasonal changes to precipitation in Denmark based on Olesen et al. (2014) and linear scaled midpoint values.**

| Change factor for seasonal precipitation (-) | Winter [CF$^{winter}$] | Spring [CF$^{spring}$] | Summer [CF$^{summer}$] | Fall [CF$^{fall}$] |
| --- | --- | --- | --- | --- |
| **Low expected change (RCP2.6)** | 1.0 | 1.0 | 1.0 | 1.0 |
| **Mean expected change** | 1.1 | 1.05 | 0.9 | 1.05 |
| **High expected change (RCP8.5)** | 1.2 | 1.1 | 0.8 | 1.1 |

### 3.2 Defining states

For Denmark the state space (Eq. 1) is defined with a total of eight states based on the expectations to climate change listed

15   in Tables 2 and 3 with four seasonal states defined for the non-extreme events and three states for the different extreme event levels:

$$E = D^{dry} \cap D^{winter} \cap D^{spring} \cap D^{summer} \cap D^{fall} \cap D^2 \cap D^{10} \cap D^{100} \text{ (Eq. 10)}$$

And correspondingly the change factors used to perturb the time series are as a starting point determined based on the mean expectations listed in Tables 2 and 3.





### 3.3 Determining state of individual events

For the determination of the state of the individual extreme events four different selection criteria are investigated, with the purpose of defining a representative return period for each event. All points mentioned refer to the events points shown in the situation depicted on Figure 2:

    A. The maximum return period is used to define the return period of the whole event (based on one point);

    B. The mean of the three largest return periods is used to define the events (based on three points);

    C. The mean of all the return periods is used to define the events (based on seven points);

    D. A selection criterion is constructed where the calculated return periods are compared to whether a predefined number of the above mentioned points are above certain regional IDF levels.

Following these selection criteria, four different systems, $S_i, i \in \{A, B, C, D\}$, are constructed and analysed.

Options $S_A$ to $S_C$ are straight forward but option $S_D$ is determined specifically for the case study. Table 4 summarizes the methodology used for option $S_D$ used in this study.

Table 5 Selection criterion $S_D$ for choosing $T_e$s at event level.

| A $T_e$ is chosen of | If | Or |
|---|---|---|
| **2 year event** | At least 4 points from the event has a return period above 0.5 years | At least 2 points from the event has a return period above 2 years |
| **10 year event** | At least 3 points from the event has a return period above 2 years | At least 2 point from the event has a return period above 10 years |
| **100 year event** | At least 3 points from the event has a return period above 10 years | At least 2 point from the event has a return period above 100 years |
| **Non-extreme event** | None of the above criteria are met | |

### 3.4 Volume correction based on seasonal dependence of extremes

In previous studies using the SVK data set it has been shown that:

1. the extreme events account for at most 25% of the total rainwater volume on an annual basis (Sørup et al., 2016b), and

20
2. the extreme events occur mostly in the summer season (Sørup et al., 2012)




Furthermore, in the summer season the excepted seasonal change (-10%) differs mostly from the expected change in extremes (+20-40%), see Table 3 and Table 2, respectively. Based on this information the seasonal change factor for non-extreme summer events has to be adjusted to reach overall changes factors reported in Table 3. We estimate a partition between non-extreme and extreme events of $\{f_{non-extreme}, f_{extreme}\} = \{0.8, 0.2\}$ and the change factor for 2-year events,

$CF^2$, is used to represent the extremes as the largest seasonal volume by far is for the more frequent extremes (Sørup et al., 2016b). In this way the change factor for summer, $CF^{summer}$, can be adjusted from its value listed in Table 4 (0.9) as:

$$CF_{adjusted}^{summer} = \frac{CF^{summer} - CF^2 f_{extreme}}{f_{non-extreme}} = \frac{0.9 - 1.2*0.2}{0.8} = 0,825. \qquad \text{(Eq. 11)}$$

In other words the change factor for non-extreme summer events are modified from -10% to -17.5% in order to compensate for the positive change of +20-40% to the extremes occurring in the summer period. For the other seasons such an

adjustment is not needed.

## 4    Results

### 4.1 Evaluation of Selection Criteria

The 10 time series are perturbed using the four different state selection criteria ($S_A$-$S_D$) and the evaluation metric is calculated using Eq. (9) with the extreme events having return periods closest to 2, 10 and 100 years (Table 6). Overall, state

selection criterion $S_D$ outperforms the other alternatives even though all selection criteria seem reasonable as all estimated deviances are below 13% of the expected changes.

**Table 6 Calculated skill scores, Φ, for the four selection criteria A-D calculated using Eq. (10).**

|   | $S_A$ | $S_B$ | $S_C$ | $S_D$ |
|---|---|---|---|---|
| **Φ** | 9.3% | 8.5% | 12% | 6.4% |

In order to study the performance for each state we construct the skill score variable of Eq. (8) and plot them against the

duration for the individual extremes and against months for seasonal precipitation (Figure 3). Plotted this way 100% represent a perfect fit, 0% represent no change and everything positive represent a change in the right direction. For the 2-year return levels both state selection criteria $S_B$ and $S_D$ perform similarly and with a relative change close to 100 %. State selection criterion $S_A$ overestimates the 2-year return level with approximately 10 % on average and state selection criterion $S_C$ likewise underestimates it, which still corresponds to a positive change for the events (Figure 3a). For the 10-year return

level, all state selection criteria perform similarly very well (Figure 3b). When the 100 year return level is evaluated the reason for criterion $S_D$'s better overall performance become clear; it is the only criterion that does not systematically underestimate this return level (Figure 3c). Even so, all criteria produce results where the direction of change is correct. Given the inherent uncertainty in estimating the actual levels of such events obtaining close to 85% of the expected change is

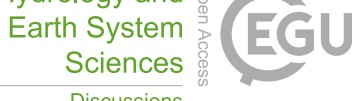



considered good. With respect to the seasonal behaviour all state selection criteria have approximately the same performance at a level close to 100% (Figure 3d).

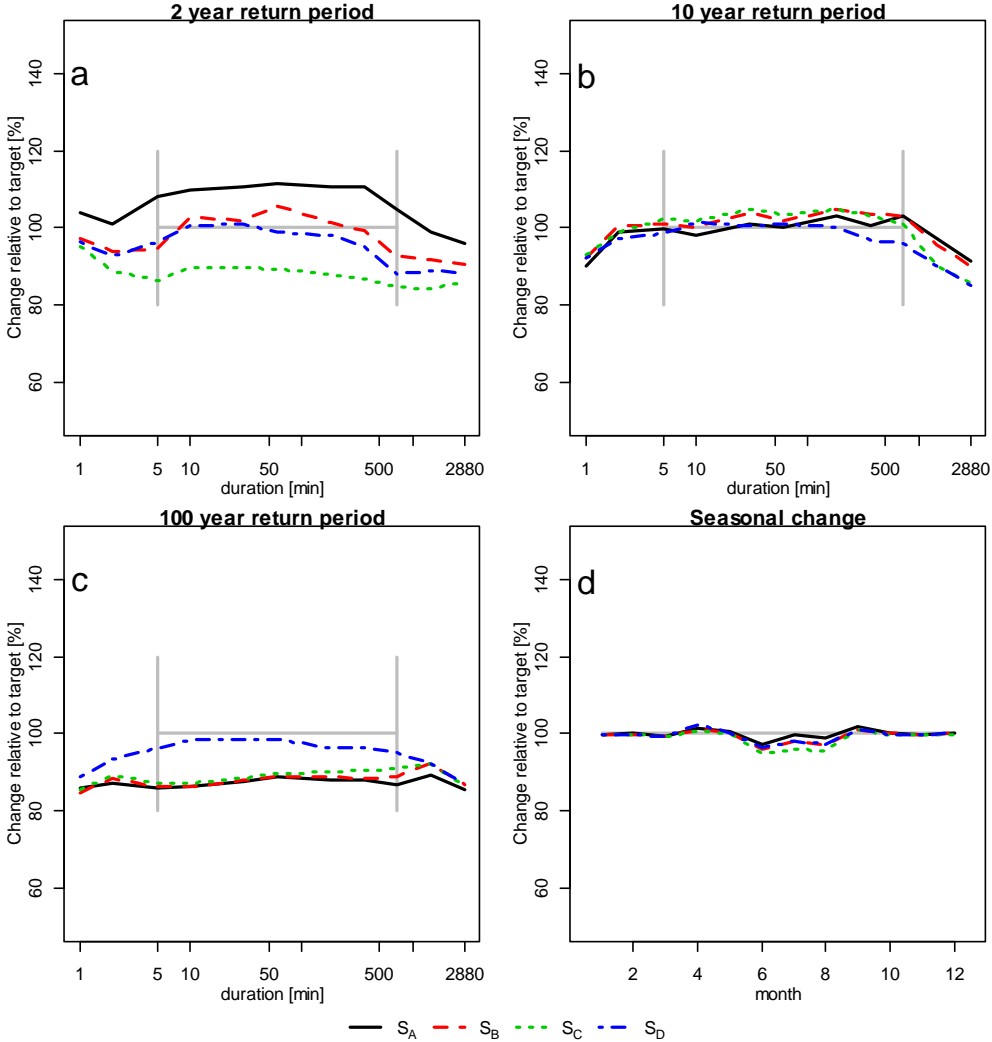

5   **Figure 3 Performance of the different selection criteria, $S_A$-$S_D$, in producing a) 2 year extremes, b) 10 year extremes, c) 100 year extremes and d) seasonal changes according to the perturbation schemes listed in Tables 2 and 3.**





The performance of all the state selection criteria drops when considering duration that are both shorter and longer than the durations used in the state selection methodology (5-720 minutes). At the minute scale this is of minor importance but at two days (2880 minutes) the tendency is very robust across different state selection criteria and extremity levels. This is most

likely because these average extreme events are caused by several events with dry periods in between. Hence the individual events are each assessed to be non-extreme and they are adjusted towards lower volumes, even though they combined are rather extreme.

### 4.2 Sensitivity Analysis with Selection Criterion D

The sensitivity analysis is carried out for the best state selection criterion only, i.e. criterion $S_D$. The resulting skill scores for

the nine individual sensitivity scenarios are listed in Table 7. The highest sensitivity is found when changing between the different extreme precipitation scenarios; with a large increase of the metric when moving from low to mean and also a notable increase when moving from mean to high scenarios. As such the performance of the methodology drops with the magnitude of the expected changes to extremes, but even for the high extremes the performance is similar to the performance of state selection criteria $S_A$ to $S_D$ in Table 6. The methodology, on the other hand, show very little sensitivity to the variation

in expectations to seasonal changes, not even for the combination where the difference between expectations to seasonal summer precipitation (-20%) and the extremes become very high (+45-100%).

**Table 7 Calculated skill scores, Φ, for selection criterion $S_D$ for the nine different sensitivity scenarios listed in Table 5 calculated using Eq. (9).**

| Φ | | Extremes | | |
|---|---|---|---|---|
| | | **Low** | **Mean** | **High** |
| **Seasonality** | **Low** | 0.0% | 6.0% | 8.6% |
| | **Mean** | 1.0% | 6.4% | 8.8% |
| | **High** | 1.2% | 6.3% | 8.8% |

For all extreme indices (Figure 4a-c) the sensitivity of the expected change of extremes is notable and especially for the 100 year return level it is clear that performance drops with increased magnitude of the expected changes to extremes (Figure 4c) but only to levels comparable to that of the state selection criteria $S_A$-$S_C$ as shown in Figure 3. Again the performance for two day events (2880 minutes) is worse than average as also seen in Figure 3. For seasonality (Figure 4d) the general picture is that the sensitivity of both expectations to seasonality and extremes are of less importance and at a similar level, which in

general is a lower level than the one observed for the three extreme indices.





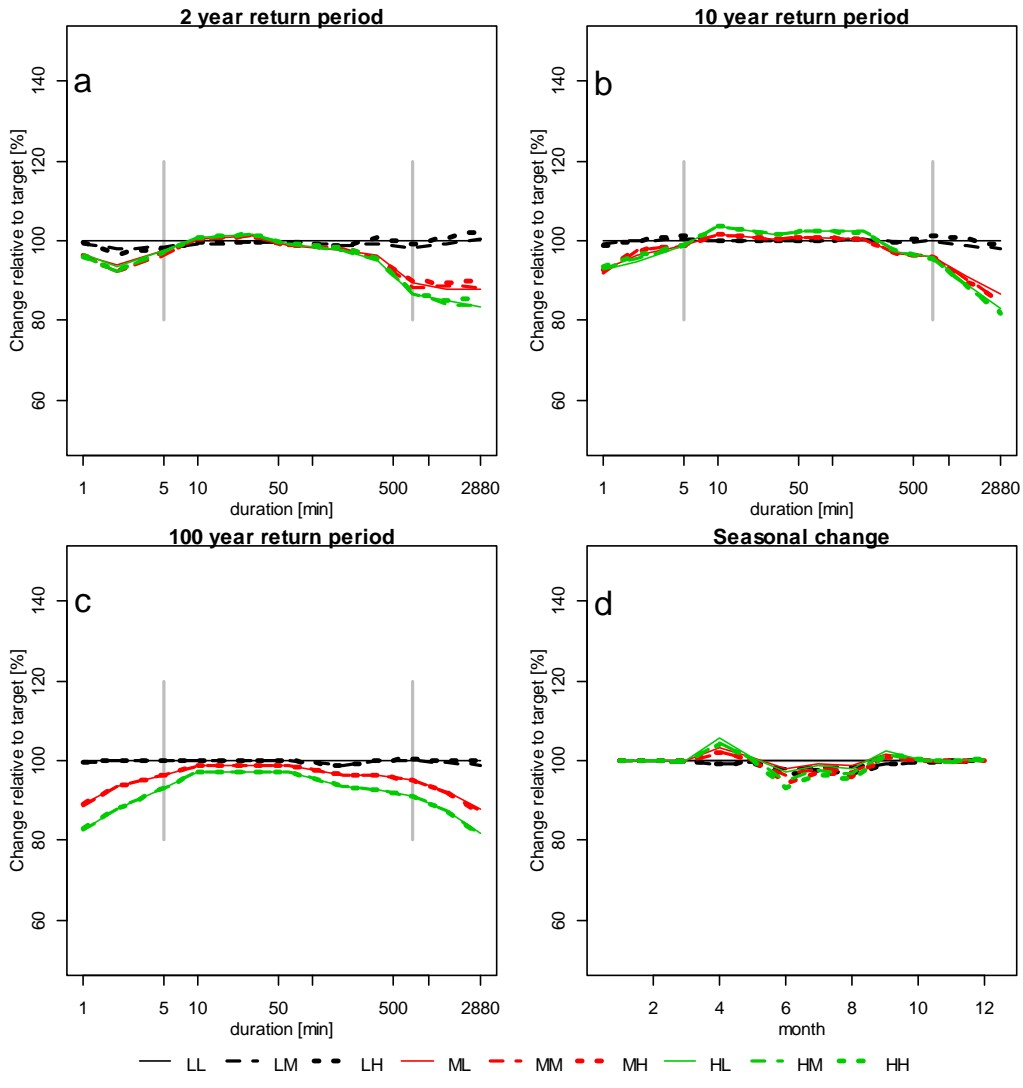

**Figure 4 Performance of selection criterion $S_D$ for different parameter values as specified in Table 5 for a) 2 year extremes, b) 10 year extremes, c) 100 year extremes and d) seasonal changes under climate change.**





## 5 Discussion

The proposed framework is very flexible and the separation of dry, non-extreme and extreme weather makes it possible to very effectively perturb time series to reflect different changes in different categories. The presented case study use eight states to distinguish between different levels of extremes and different season and is able to produce time series that

satisfactory represent the expected changes listed in Tables 2 and 3. For other places a different number of states could be relevant and the seasonal partition could be different depending on the local climate and expectation to climate change. The proposed modelling framework fully supports this.

Four different state selection criteria over specified event durations are tested in the present study, see Section 2.2, as these covered realistic possibilities for the data set used in this study and the focus on urban hydrology. As such, different state

selection criteria for different event durations could be relevant in different contexts and could, as illustrated by state selection criterion $S_D$, be specified as very subjective and case specific criteria. In this study the subjective state selection criterion $S_D$ outperforms the other criteria, see Table 6 and Figure 1, but the superiority is mainly due to its ability to produce the largest changes for the very large, and very uncertain, extreme events. If this part of the evaluation is disregarded criteria $S_B$ and $S_D$ has very similar performance pointing at criterion $S_B$ as being a good onset for investigating data sets where no

presumptions exist and no case specific criterion can be constructed.

All state selection criteria showed a drop in performance for longer duration events than the ones used in the methodology; this is likely due to the used event definition with a minimum of 60 minutes of dry weather between individual events which will mean that very long lasting extremes likely are split into several events and therefore not identified as extremes. A different event definition with longer minimum dry period between events could probably partly solve this, but it would

reduce the number of events markedly and increase the chance of small events close to extremes being seen as part of the extreme with a somewhat false classification as a consequence.

The methodology is somewhat sensitive to the magnitude of the perturbation factors, see Section 4.2, but the sensitivity is not very dominant and is only at the same size as the sensitivity of the different state selection criteria. Also, the methodology does not address the possibilities of changes to dry spells or changes to the occurrence rate of extremes in

general. A future research direction could be to study how the state selection criteria along with the semi-Markov system applied here can be used to generate fully stochastic time series where both the inter-event time and the occurrence probability of the extreme states will be included as criteria that can be changed to meet the expectations to climate change.

## 6 Conclusions

The proposed methodology is a promising way of creating artificial perturbed precipitation time series, which can represent a

changed climate and be used as input in hydrologic and hydraulic models. The methodology perturbs existing time series based on a semi-Markov system where precipitation time series are split into events characterized as *dry*, *extreme* or *non-extreme*. The wet events are divided into different states based on an Intensity-Duration-Frequency relationship based state



selection criterion. Of the four tested state selection criteria, the case specific show the best results, but also the more general criteria could be of use when less knowledge about the precipitation regime is available. The sensitivity of the methodology was tested against very different expectations to climate change both with respect to seasonal changes and changes to extremes and is generally very robust, also regarding seasons where the general change is negative while the expectations to extremes is positive. The produced time series satisfactory reproduce changes across all seasons and across all levels of extremes relevant for urban hydrology.

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
