# Peer review of "Formulating and testing a method for perturbing precipitation time series to reflect anticipated climatic changes"

_Hydrology and Earth System Sciences, 2016_

## Referee Comment (RC1) · Anonymous Referee #1 · 7 Nov 2016

**General comments**

In the paper the authors propose a method for perturbation of observed sub-daily precipitation time series in order to result in the same changes as given by a climate change scenario. The method is simple, i.e. the observed rainfall events are classified into several (eight in the paper) states and each state is perturbed by its specific change factor. Further the method assumes that extreme precipitation occurs only during summer (but can be easily modified to allow for extremes in different seasons). The method is evaluated on precipitation from 10 stations across Denmark and climate change scenarios of rainfall extremes and seasonal precipitation.

The method can be potentially used in many practical applications, the paper is in

general well written, yet there are some parts needing clarification. In addition I believe the paper could benefit from discussion of alternative approaches, which is not present at all.

**Specific comments**

1. Though the idea is very simple, the presentation is rather formal and sometimes difficult to follow. I understand that the authors like to present their method correctly, however, I wonder whether it would not be clearer to the reader to avoid formal definition with state spaces, semi-Markov systems, discrete-time counting processes etc. especially when there is no explicit use made from it and the whole trick is in class-based multiplication.

2. The difficult part in the application of the proposed methodology is the identification of event state. This is because there is a mismatch between event definition (based on minimum inter-event time) and definition of climate change scenario (based on maximum intensity). Apparently a single event may include precipitation with different return levels at different time scales. Authors therefore test several approaches to determine the class of the event and evaluate the skill of each variant. Since the determination of event return period is one of the key parts of the study I suggest to formulate the problem more explicitly. As the paper reads now, it is somewhat hidden.

3. I am quite confused with eq. 8, its application in figures 3 and 4, tables 6 and 7 and related text: eq. 8 defines the average percentwise difference between the perturbed return levels $z*$ and their expected values $z$ as $\Phi = 100(1 - -z*/z)$, resulting in 0 if there is perfect fit, negative values if the perturbed changes are larger than expected and positive values otherwise. In contrast to this, authors state in description of fig. 3 showing $\Phi$ against duration (p. 11, l. 20) that "100

4. Authors are trying to determine the state of an event on the basis of the return level of precipitation at several (7?) time-scales. Are there (/have authors tried) any other options? Could, for instance, the event change factor be derived from some weighted

combination of relevant factors similarly as in eq.11? Or would not be in general more feasible to define climate change scenarios on the basis of the changes in rain event characteristics (see e.g. Svoboda et al., 2016a, b)?

5. The change factors in Tab. 3 are duration-independent, is it reasonable assumption? Are the changes (after perturbing the observed data) also independent on duration? Could you provide some information, how the values presented in Tab. 3 relate to CORDEX projections?

6. Authors are averaging seasonal changes of RCP2.6 ("low change") and RCP8.5 ("high change") in order to get "mean change" scenario and state that this is in order to correspond to the "average A1B emission scenario". While the A1B was the most frequently used scenario, it is not clear, why it should be average scenario. In fact, e.g. considering the radiative forcing, the A1B scenario is somewhere close to the upper third between RCP2.6 and RCP8.5 in the end of the 21st century. Please find different arguments for the setup of your experiment.

**Technical corrections**

p. 1, l. 25 – "Climate change impact water management" - change to "Climate change impacts water management" p. 2, l. 8 – delete "sought" between "often" and "solved" p. 3, l. 5 – please define || in |E| p. 3, l. 17 – remove "the" between "that" and "there" p. 5, l. 8 – insert "events" between "between" and "of" p. 5, l. 13 – change "is" to "are" in "extremes is identified" p. 6, l. 26-29 – please try to write the sentence in standard way p. 7, l. 3 – please specify, how $z*$ is calculated p. 8, l. 3 – please insert "in" between "used" and "this" p. 8, l. 8 – is the reference to submitted paper (from 2009) correct? p. 10, point D – please rephrase

**References**:

Svoboda, V. et al. (2016a) Projected changes of rainfall event characteristics for the Czech Republic. Journal of Hydrology and Hydromechanics, 64(4), DOI: 10.1515/johh-

2016-0036

Svoboda, V et al. (2016b) Characteristics of rainfall events in RCM simulations for the Czech Republic. Hydrology and Earth System Sciences Discussions, in revision

---

## Referee Comment (RC2) · Anonymous Referee #2 · 1 Dec 2016

MS Title: Formulating and testing a method for perturbing precipitation time series to reflect anticipated climatic changes

Authors: Hjalte Jomo Danielsen Sørup et al.

MS No.: hess-2016-500

MS Type: Research article

Special Issue: Rainfall and urban hydrology

In this paper the authors introduce a methodology to perturb historical precipitation time series at the minute scale to model non-linear expectations of climate change.

[Figure]

This is an innovative method because very few previous research studies have dealt with such fine resolution time series. One of the important limitations of the proposed approach is that it is a deterministic model that may be questionable under climate change scenarios.

The literature review presented in this manuscript could be improved. A more specific literature review should also be included in the introduction section. In particular, the cited studies do not adequately support the proposed methodology. The method presented could also be presented more clearly. Perhaps a flow diagram in the methodology section would add clarity.

The authors have categorized extreme events based on 2, 10 and 100-year return periods. It would be interesting to include other categories between 10 and 100 years and at least a 50-year return period should be included, as there is a big difference between a 10 and 100 year return period event.

As per the authors' description (page 2, lines 11 – 17), the rainfall time series for a changed future climate is generated by perturbing expectations of future precipitation into the observed time series. The authors should perhaps explain how they propose to incorporate persistence in the generated future series because persistence is important when precipitation series are used in a hydrological model.

The authors claim that they have generated time series for future changed climates, and so it would be interesting to see some time series figures and statistics to show that these are indeed realistic.

On what basis do the authors set the selection criteria to determine the states of each event as listed in Table 5, particularly for the 100 year events? Why did they use three/two points? This selection should be explained in more detail. Perhaps include references that justify this selection process or explain why a subjective approach was taken?

[Figure]

The following two references are included in this manuscript and seem to be quite important because the authors use information from those references (expected change factor listed in Tables 3 and 4 and for the perturbation analysis). But unfortunately, these references were not accessible and/or were in another language and so it was difficult to look into these aspects in more detail:

1. Gregersen, I.B. Madsen, H., Linde, J.J. and Arnbjerg-Nielsen, K.: Opdaterede kl 5 imafaktorer og dimensionsgivende regnintensiteter (Updated climate factors and design rain intensities) - Spildevandskomiteen, Skrift nr. 30. The Danish Water and Wastewater Committee under the Danish Engineering Society, Copenhagen, Denmark. In Danish. 2014.

2. Olsen, M., Madsen, K.S., Ludwigsen, C.A, Boberg, F., Christensen, T., Cappelen, J., Christensen, O.B., Andersen, K.K. and Christensen, J.H.: Fremtidige klimaforandringer i Danmark 5 (Future climate changes in Denmark). Danmarks Klimacenter rapport nr. 6 2014. Danish Meteorological Institute, Copenhagen, Denmark. In Danish. 2014.

Either alternative references should be provided or if this is not possible then more detail on the information used should be provided.

Minor corrections:

1. The statement on page 1, line 31 is not always true. There have been research studies undertaken where streamflow directly is downscaled from GCMs to assess the future changes due to a warmer climate without using hydrological models.

2. In section 2.1 Modelling Framework (page 3, line 6), the meaning of lowercase "d" is not clear. Please explain "d".

3. The manuscript needs more careful proofreading, i.e. the reference to Table 4 on page 10 should be corrected to Table 5.

4. The English and grammar also need further attention, i.e. straight forward should read straightforward on page 10.

5. "Has" should read "have" in several instances in Table 5.

---

## Author Response (AR1)

**Table of Content**

**Reply to Anonymous Referee #1**

We greatly appreciate the review and acknowledge that the comments and suggestions will lead to an improved paper.

**Regarding the general comments**

It is not assumed that extreme precipitation only occur in summer even though the majority of the events occur during this season. The summer season is particularly interesting because most extremes occur here and will increase further while in general the precipitation amounts are decreasing. In other seasons there is no such difference between average and extreme properties of the changes.

Regarding a discussion of alternative approaches we will add further discussion of the possibilities beyond the presented approach (also facilitated by the comments from Referee #2) that could also include alternative approaches to reach the same endpoint. We will focus on using Markov models for precipitation and the two downscaling approaches Delta Change and Distribution Based Scaling, which has inspired us in defining the framework.

**Regarding specific comments**

In 1. a point is raised that the semi-Markov system used to frame the approach is "rather formal" given that essentially the approach is very simple. However, the use of a somewhat extravagant terminology has advantages if the model should be extended into a stochastic formulation. The application of a semi-Markov system for setting up different numbers of classifications is straightforward and extending the system to a stochastic model on a more general level is possible.

As pointed out in 2. the difficulties in assigning a single event state is central to the approach and the section (Section 3.3) will be extended with an elaboration of the mathematical considerations as well as the importance of these.

Regarding 3. we thank the reviewer for pointing out the error in our manuscript and will correct it in the final version of the paper.

As stated in 4. there could be other ways to determine the return level of the individual events. We will try to make this section clearer, especially since both reviewers point out that the current manuscript is unclear here. We will focus on the need to test the approach in relation to how it will be used and that users can and should define suitable metrics depending on the actual use of the constructed series. The defined metrics was chosen because it is a basic requirement that the series should be able to fulfil these criteria before they are used in Denmark (which other approaches have failed).

The duration independence of the used change factors (as raised in 5.) is based on the official recommendations for Denmark. We agree that it is probably a bad assumption. One of the justifications for choosing this approach is that often climate change impacts are based on design storms which makes duration-specific change factors difficult to employ. It might be an option to identify duration specific change factors and use them within the presented framework. However, it would probably require some further analyses of the structure of events which goes beyond the current study.

The point raised in 6. about the A1B scenario's relative place in relation to RCP2.6 and RCP8.5 is much
appreciated and really help demonstrate the difficulties of working with derived data based on different
generations of climate model scenarios. The idea has not been to indicate that the A1B scenario was the
midpoint between the two RCP scenarios, but merely to state that it was somewhere in between. Also, as
illustrated by the results in Figure 4, we use the notion of "low", "middle" and "high" emission scenarios in
an assessment effort towards documenting the sensitivity of the approach towards the absolute magnitude
of the expectations to climate change. We will alter the relevant sections to make this clearer.

**Regarding technical corrections**

We will make the grammar corrections and ad explanation for || and $z*$ as asked for. As for the reference
"Madsen et al. (2009, submitted)" we will rewrite the sentence to highlight that what is referred to is
Madsen et al. (2009) and Madsen et al. (submitted) where the last one is an update of the model described
in the first one.

**Reply to Anonymous Referee #2**

We very much appreciate the review and acknowledge that the comments and suggestions will lead to an improved paper.

60   The first concern raised regards the fact that the approach is deterministic. We agree with this concern because there are features of climatic changes that will be difficult to implement in a deterministic framework. Our focus has been on making a proof-of-concept of the methodology by testing a method currently applied on daily rainfall (Delta Change, *DC*, and Distribution Based Scaling, *DBS*) to much higher resolution. We have formulated the framework as a semi-Markov process in order to be able to extend it
65   into a stochastic framework.

We are happy to extend the literature review . We would suggest to cover in more detail the use of Markov models in hydrological applications as well as a more in-depth presentation of the *DC* and *DBS* methods thereby setting a better foundation for the present approach. This could, indeed, be further supported by a flow diagram in the methodology section.

70   The 2, 10 and 100-year return periods used for the Danish case study are included as these are the categories for which climate change predictions exist. These return periods correspond to the typical uses of extreme precipitation for pipe flow capacity, surcharging, and flood risk management, respectively. In principle we would prefer to use a smaller return period, e.g. 50 years, because of the relative short time series used in the study. However, we have chosen the return periods for which official recommendations
75   exist and prefer to keep it that way. We note that because of the correlation introduced when ranking extreme series the 50 year event is implicitly covered by the chosen return periods and that the method can easily be adapted to other return periods.

Regarding persistence of the time series, the presented approach retains the present day time series characteristics both when it comes to intra and inter events persistence. For the intra event persistence,
80   this is believed to be the best available option and indeed a standard assumption in most down-scaling approaches to yield precipitation series with sub-hourly resolution. For the inter event persistence, this is not an ideal approach, as some RCMs predict changes for these statistics. This is, however, a highly debated topic as the regional climate models do not agree on these parameters for the case study location (Boberg, 2010). In a future extension into a stochastic framework, we agree that this is one of the very important
85   factors that have to be modelled specifically to further enhance the methodology.

As the generated time series for future climate maintain the structure of the present day time series, the expectation metrics calculated and reported in Figure 3 and other places are really the best way to show that the future time series are realistic with respect to the perturbation we apply in the approach.

With respect to the selection criteria reported in Section 3.3 and specifically in Table 5, we agree that the
90   rationale behind the choices should be elaborated both with respect to mathematical description (as pointed out by Referee #1) as well with thorough explanations of the subjective choices made for option D (Table 5).

With respect to the two references in Danish that are used as references for the expectations to climate change in Denmark, we will look into if there are international literature that has come out based on these

95     reports and, if that is not the case, add more detailed referencing (for instance the figures reported in Table 3 are based on the results reported in Tabel 1 of Gregersen et al., 2014) along with a more general presentation of the reports.

We will make changes to the paper that accommodate the more technical comments by the reviewer, including careful proofreading.

100 ## Additional reference

Boberg F (2010): Weighted scenario temperature and precipitation changes for Denmark using probability density functions for ENSEMBLES regional climate models. Danish Climate Centre Report 10-03. https://www.dmi.dk/fileadmin/Rapporter/DKC/dkc10-03.pdf.

**List of changes**

p2ll10-18: paragraph added on use of Markov models in hydrological applications

Pp2-3ll27-1: paragraph added explaining Delta Change and Distribution Based Scaling in more detail

p3ll10-12: Flow diagram added

p3l18: clarification of $|E|$

p3l21: clarification of the meaning of $d_1$ and $d_2$

p6ll2-9: paragraph added elaborating on how to define events

p8ll12-15: revision of sentence as suggested by reviewer

p8ll26: correction of equation for calculating $\Phi$ as pointed out by reviewer

p9l20: clarification of difference between Madsen et al. (2009) and Madsen et al. (In Review)

p10ll9-13: elaboration of the report by Gregersen et al. (2014) and the results used from this report

p11ll1-7: elaboration of the report by Olesen et al. (2014) and the results used from this report

p11ll10-11: clarification of the relationship between SRES and RCP scanarios

p12ll3-26: elaboration and extension of the methodology used to determine the return period of individual events including mathematical expressions to level this section with the rest of the methodology section

Furthermore, a lot of small textual corrections and updates of Figure and Table numbers have been done throughout the text. New references has been added to support the new paragraphs and direct links to the two Danish reports used has been added to the references.

[revised manuscript text omitted]